# Direct and Indirect Evidence of Effects of *Bacteroides* spp. on Obesity and Inflammation

**DOI:** 10.3390/ijms25010438

**Published:** 2023-12-28

**Authors:** Liangliang Wu, Seo-Hyun Park, Hojun Kim

**Affiliations:** 1Department of Rehabilitation Medicine of Korean Medicine, Ilsan Hospital of Dongguk University, Goyang 10326, Republic of Korea; wuliangliang@dgu.ac.kr; 2Department of Rehabilitation Medicine of Korean Medicine, Bundang Hospital of Dongguk University, Seongnam 13601, Republic of Korea; np190318@gwmail.dongguk.edu

**Keywords:** *Bacteroides* spp., obesity, metabolic diseases, probiotics, inflammation

## Abstract

Metabolic disorders present a significant public health challenge globally. The intricate relationship between the gut microbiome, particularly *Bacteroides* spp. (BAC), and obesity, including their specific metabolic functions, remains partly unresolved. This review consolidates current research on BAC’s role in obesity and lipid metabolism, with three objectives: (1) To summarize the gut microbiota’s impact on obesity; (2) To assess BAC’s efficacy in obesity intervention; (3) To explore BAC’s mechanisms in obesity and lipid metabolism management. This review critically examines the role of BAC in obesity, integrating findings from clinical and preclinical studies. We highlight the changes in BAC diversity and concentration following successful obesity treatment and discuss the notable differences in BAC characteristics among individuals with varying obesity levels. Furthermore, we review recent preclinical studies demonstrating the potential of BAC in ameliorating obesity and related inflammatory conditions, providing detailed insights into the methodologies of these in vivo experiments. Additionally, certain BAC-derived metabolites have been shown to be involved in the regulation of host lipid metabolism-related pathways. The enhanced TNF production by dendritic cells following BAC administration, in response to LPS, also positions BAC as a potential adjunctive therapy in obesity management.

## 1. Introduction

The human gastrointestinal tract is a dense microbial habitat hosting approximately 4 × 10^13^ bacteria, predominantly *Firmicutes*, *Bacteroidetes*, *Proteobacteria*, *Actinobacteria*, and *Verrucomicrobia* [1,2]. *Firmicutes* and *Bacteroidetes* are dominant constituents of the gut microbiota [3]. *Bacteroides* (BAC), a prominent genus within the *Bacteroidetes* phylum, is characterized by anaerobic, bile-resistant, non-spore-forming, Gram-negative rods [4]. Significant taxonomic revisions in recent decades have expanded the genus to encompass over 20 distinct species [5,6]. Prominent members of BAC in the colon include *B. vulgatus*, *B. distasonis*, and *B. thetaiotaomicron* (at levels of 10^10^ per g dry weight of feces); species such as *B. fragilis*, *B. ovatus*, *B. eggerthii*, *B. uniforms*, and *B. “B5-21”* are less abundant but still present in substantial concentrations (10^9^ per g dry weight of feces). BAC are integral to the ecological balance of the colon [7,8]. Their robust adaptability within the host and potential benefits have garnered significant attention from researchers, positioning them as valuable models or communities for studies in the realm of gut microbiology.

Recent studies have highlighted the potential of prebiotic and probiotic supplementation for weight management, with negligible adverse effects [9]. For instance, supplementation with *Lactobacilli* (phylum *Firmicutes*) has exhibited favorable effects on blood glucose levels, lipid profiles, weight, and body fat in overweight adults and those with obesity, as corroborated by several clinical studies [10,11]. *Bifidobacterium* spp. has also effectively reduced fat and body weight in preclinical and clinical studies of obesity and type 2 diabetes [12,13]. Furthermore, BAC have been shown to activate fat oxidation in adipose tissue via the bile acid-TGR5-PPARα axis, thus increasing energy consumption [14]. A previous review also emphasized the ability of BAC to mitigate various diseases, such as *Helicobacter hepaticus* infection, oxazolone-induced experimental colitis, *Vibrio parahaemolyticus* infection, antibiotic-associated diarrhea, high-fat diet-induced cardiovascular disease, and cancer [15]. BAC-derived compounds and metabolites interact with various immune cells and contribute to immune response activation. Notably, Polysaccharide A (PSA), derived from the human commensal *B. fragilis* NCTC9343, is a symbiotic factor that stimulates immune development in mammalian hosts, stimulates dendritic cell maturation and T-cell proliferation, in addition to possessing abscess-inducing properties [16,17].

BAC engage in both competitive and symbiotic interactions with adjacent bacterial species, thereby markedly influencing gut microbiota dynamics. BAC employ two distinct strategies for polysaccharide breakdown: they either prevent other bacteria from accessing the products of polysaccharide degradation or make these products available to other species. For example, *B. thetaiotaomicron* exhibits limited degradation of the yeast cell wall polysaccharide α-mannan outside the cell, transporting the relatively large α-mannan polysaccharides into the periplasm for further breakdown into manno-oligosaccharides and mannose [18]. This strategy discourages the utilization of partially degraded α-mannan by bacterial species lacking the requisite transporter for this unique polysaccharide, thereby favoring the proliferation of species equipped to utilize it [19]. However, previous research records show that BAC is identical to some commercial conventional probiotics [20,21,22]. The health-promoting properties also depend on the strain. Although the PSA produced by non-enterotoxigenic *B. fragilis* attenuates infections and aids colonization, it also serves as a symbiont for bacterial growth [23,24]. Despite the diversity of research directions, there is a common thread among numerous previous studies that have focused on BAC, with most findings supporting their beneficial effects against obesity, metabolic disorders, and inflammation.

Recent investigations have elucidated that BAC exhibited anti-inflammatory properties within the body [25]. Notably, the interplay between obesity and inflammation is well-documented, with both conditions often presenting concurrently [26]. Despite these findings, the direct impact of BAC in mitigating obesity in humans has not been thoroughly explored, underscoring the need for comprehensive research in this area.

In our analysis, we primarily referenced both direct and indirect experiments in previous studies that explored the effects of BAC on weight loss, encompassing animal models and human clinical trials. Given the absence of clinical trials that directly elucidate the mechanisms whereby BAC supplementation modulates obesity in humans, we aimed to extrapolate the potential mechanisms by summarizing the outcomes of preclinical and clinical studies associated with increased concentration of BAC in the gut. Our review included an exploration of the efficacy and potential mechanisms of BAC related to its anti-obesity and lipid-reducing properties, encompassing the impact of BAC on the structure and diversity of the host gut microbiota as well as the potential involvement of BAC-derived metabolites in the regulation of host metabolic signaling pathways. This analysis is pivotal for advancing targeted obesity interventions.

## 2. Materials and Methods

### 2.1. Data Source and Search Strategy

We conducted a comprehensive literature search in November 2022 using the PubMed, Google Scholar, Cochrane, Scopus, and Web of Science databases within a restricted time period between 1 January 2000 and 30 November 2022. We employed specific search strings combining terms and Boolean operators, such as “*Bacteroides* AND obesity AND trial”. These terms were adapted for each literature database and combined with database-specific filters for human clinical trials and preclinical trials in animals.

### 2.2. Inclusion and Exclusion Criteria

We searched for studies demonstrating that BAC effectively treated obesity, metabolic diseases, and inflammation. Because of the lack of studies that focused on the efficacy of BAC for obesity, the searched articles included both clinical trials and preclinical studies using animal models. However, the screening and selection were conducted separately, according to the below criteria. There were no language restrictions, and only the literature with abstracts was reviewed. In addition, studies with incorrect BAC descriptions or those without statistical analyses were excluded. 

The inclusion criteria for clinical trials with indirect evidence were as follows: (1) human studies wherein participants were clearly defined as being overweight or obese at baseline; (2) participants who were overweight or obese and had a significant reduction in body weight, body fat, and other relevant body metrics after treatment; and (3) the richness, diversity, abundance, and diversity of BAC changed as the body metrics gradually normalized. The exclusion criteria were as follows: (1) unavailability of the full text; (2) no weight loss with treatment; (3) animal models; and (4) failure to provide results related to BAC. For the article type, it was not limited to randomized controlled trials. Observational studies, which compared the obese characteristics to healthy subjects’ characteristics, were also included. In the case of intervention, any reasonable treatments for obesity were included, and there was no limitation for the comparator. The before–after studies for a single group were also included; however, reviews, case series, or case reports were excluded.

The inclusion criteria for preclinical trials with direct certification were as follows: (1) animal model; (2) involvement of a control group; and (3) treatment or intervention with BAC supplementation. The following exclusion criteria were used: (1) not an original paper; (2) lack of comparison; and (3) not using animal models such as human studies; no direct results demonstrated that BAC exhibited anti-obesity effects. The key data for inclusion are summarized in Figure 1.

### 2.3. Data Extraction and Data Analysis

In the initial phase of our research, we amalgamated the search results sourced from various databases to compile a comprehensive dataset. Subsequently, this dataset underwent a rigorous process of de-duplication, ensuring the removal of any redundant entries. The task of evaluating and determining the relevance of the collated data was undertaken collaboratively by a trio of researchers. In cases where differing viewpoints emerged, these discrepancies were diligently addressed through in-depth discussions, leading to a consensus. This methodical approach not only ensured the accuracy and reliability of our data selection but also underscored the collaborative integrity of our research methodology.

From each selected publication, data extraction was performed separately based on the characteristics of the studies. 

For the clinical trials used for indirect proof, the following data were extracted: information about sex; age; physical data related to obesity; and the changes caused by BAC were obtained for human participants. For the preclinical studies, information on study design, interventions, comparisons, strains and doses, intervention periods, and primary outcomes were extracted.

The primary outcome indicator for indirect evidence in clinical trials was a change in the abundance of BAC, whereas the primary outcome indicators for direct evidence in preclinical studies were body weight, liver weight, body fat, blood lipids, liver injury, steatosis, hepatic lipid metabolism, and insulin levels.

For studies utilizing BAC in combination therapies, a sub-analysis was conducted to isolate the effects attributable to BAC alone based on available data. Studies that indicated improved health markers without weight loss were subjected to a separate analytical process to determine the potential role of BAC in metabolic health. 

Given the established anti-inflammatory properties of BAC in human subjects, this study aimed to explore the broader implications of BAC on obesity management. The data analysis is, therefore, organized into four distinct sections: The first part comprises studies that indirectly demonstrated changes in BAC after treatment with probiotics, herbs, dietary supplements, and synthetic probiotic supplements, either alone or in combination with other treatments, resulting in weight loss in overweight human participants and patients with obesity using medical subject terms (*Bacteroides* spp.). The second section explores the differences in the abundance of BAC in the gut microbiota of individuals with obesity and healthy individuals using medical subject headings (*Bacteroides* spp., gut microbiota, clinical trials, overweight, obesity, and 16S rRNA gene sequencing). Because there are no clinical trials of BAC directly treating obesity, the third part comprises information regarding preclinical studies demonstrating that BAC can treat obesity using medical subject headings (*Bacteroides* spp., mice, preclinical studies, overweight, obesity, weight loss). The fourth section comprises preclinical wherein BAC were found to directly affect inflammation (*Bacteroides* spp., mice, preclinical studies, inflammation). 

### 2.4. Quality Assessment

We adapted the Downs and Black checklist to suit the nuances of the reviewed studies, focusing on parameters critical for assessing the reliability and applicability of obesity and microbe-related research [27]. The checklist contained 27 items across the following five domains: reporting; external validity; internal validity (bias); internal validity (confounding); and efficacy. After assessment, the studies were categorized according to their scores as excellent (25–27), good (20–24), fair (15–19), or poor (<14). If no patients were lost to follow-up, the ninth analyzed indicator was deleted, and an extra point was added to the scoring results. The quality of the included studies was presented using Review Manager version 5.3. (Copenhagen, the Nordic Cochrane Centre, the Cochrane Collaboration, 2014).

## 3. Results

### 3.1. Study Description

The primary objective of our study was to investigate the impact of BAC on human obesity. Our research commenced with a systematic search for clinical trials specifically addressing the direct treatment of obesity using BAC strains. Regrettably, no relevant clinical trials were identified. However, we identified pertinent preclinical trials and collected clinical trials for indirect evidence. We curated clinical trials where BAC strains indirectly influenced obesity treatment. Of the 31 relevant studies, 19 reported that interventions leading to weight loss in human participants resulted in increased BAC abundance. These findings imply a correlation between BAC and weight management, although the directionality of this relationship requires further investigation. Interestingly, two studies documented weight loss concurrent with a decrease in BAC abundance, suggesting that other unidentified factors might influence the interaction between the gut microbiota and weight regulation or that different BAC strains might have varying effects. The ten comparative microbiota analyses across individuals with different demographic backgrounds and body compositions further improved our understanding of the role of BAC. Although they did not provide direct evidence of weight loss, they contributed significant insights into the natural variance of BAC in the human gut and its correlation with obesity, underscoring the need for personalized approaches to microbiota-targeted treatments. The key data for inclusion are summarized in Table 1, Table 2 and Table 3. 

The direct evidence analysis identified 142 potentially relevant articles, of which 12 met the inclusion criteria. Six of these studies, wherein BAC were directly administered via gavage, reported a consistent decrease in body weight, suggesting the potential anti-obesity effect of these bacterial strains. The more details are presented in Table 4. 

The data summarized in Table 5 from relevant studies indicate that BAC, in combination with various pharmacological agents, contribute to reducing the inflammatory response in experimental animal models. This section always reflects the mechanisms by which obesity, inflammation, and BAC may be related. In this section, only clinical weight loss trials after pharmacological interventions evaluate the specific drug of the intervention, the number of patients involved in the problem, the type of intervention, and the duration. The second part of the review considers only articles reporting fat reduction upon direct gavage of BAC strains were considered, and the types of interventions considered included specific strains, types of experimental animals, types of interventions, and duration of interventions.

### 3.2. Indirect Evidence of the Anti-Obesity Effect of BAC from Clinical Studies

Among the 18 included clinical trials, the shortest duration of intervention was 29 days, and the longest was 16 weeks. There were 13 cases wherein weight, waist circumference (WC), or body mass index (BMI) achieved significant reduction after different interventions, such as probiotics, herbs, and low-carbohydrate diets [28,29,30,31,32,33,34,35,36,37,38,39,40]; additionally, the richness and diversity of BAC strains increased to varying degrees with a significant reduction in weight [28,30,31,32,34,36,37,38,39,40,41,42,43,44]. In two studies, although no significant changes in obesity-related parameters were induced, both studies showed a decreasing trend; the intervention in both cases increased the abundance of *BAC* spp. [29,33].

A clinical trial conducted on Italian patients with obesity, wherein most of the included patients had metabolic dysfunction, showed a statistically significant reduction in body weight and adiposity after the administration of a healthy diet based on caloric intake, macronutrient composition, and fiber intake. There was a statistically significant increase in the abundance of *BAC* spp., especially *B. uniformis* (*p* = 0.005) [35]. In another study, nine healthy volunteers were selected to fast for approximately 17 h per day for 29 days; at the end of the trial, there were no significant changes in obesity-related values, but a significant increase in the abundance of both *A. muciniphila* (*p* = 0.0039) and *B. fragilis* (*p* = 0.0078) was observed [45]. Notably, an increase in the abundance of *BAC* spp. was reported in all these studies (Table 1).

However, a few clinical trials have yielded conflicting results in that obesity improved while *BAC* spp. abundance decreased. Furthermore, two clinical trials in children and women with obesity showed a decrease in *B. vulgatus* abundance after the administration of probiotics [46,47] (Table 2).

**Table 1 ijms-25-00438-t001:** Indirect evidence of the anti-obesity effect of *BAC*.

Research Purpose	Numbers of Patients Randomized/ Analyzed	Intervention Type (Regimen)	Treatment Period	Main Outcome Measure	Main Conclusion	Reference
Exploration of the anti-obesity efficacy of steamed Rehmannia glutinosa root.	Middle-aged female participants with obesity (40–65 years) 12/12	Nor group (*n* = 25) LCD group (*n* = 26)	8 weeks	BAC↑ Waist circumference *Roseburia*↑ *Dorea*↓	BAC levels are negatively correlated with waist circumference.	[28]
Assessment of metabolic markers related to obesity post-Schisandra chinensis fruit administration	Participants with obesity 40/28	SCF group (*n* = 13) Placebo group (*n* = 15)	12 weeks	BAC↑ *Akkermansia*↑ *Roseburia*↑ *Prevotella*↑ *Bifidobacterium*↑ *Ruminococcus*↓ Relationship with BAC: Fat mass	BAC levels are negatively correlated with fat mass and aspartate aminotransferase and/or alanine aminotransferase levels.	[29]
Examination of the correlation between gut microbial modulation and plasma lipopolysaccharide-binding protein (LBP) levels.	Participants who were overweight and obese 49/49	Pomegranate extract (PE) Placebo	3 weeks	BAC↑ Plasma LBP High-sensitivity C-reactive *protein* *Faecalibacterium*↑ *Butyricicoccus*↑ *Odoribacter*↑ *Butyricimonas*↑	BAC levels are positively correlated with the levels of other beneficial bacteria and health indicators.	[30]
Determination of the effects of metformin on anthropometry and gut microbiota in non-diabetic women with obesity under a low-calorie regimen.	Women with obesity 46/36	Metformin + LCD group (*n* = 20) Placebo + LCD group (*n* = 16)	8 weeks	BAC↑ BMI Insulin concentration *Escherichia*/*Shigella* abundance *Roseburia*↑ *Blautia*↑ *Butyrivibrio*↑	BAC levels are negatively correlated with BMI and insulin concentrations.	[31]
Evaluation of the role of daily avocado intake within a hypocaloric diet on weight management, inflammation biomarkers, and gut microbiota diversity.	Healthy men and women who were overweight/obese 63/51	Avocado daily intake (*n* = 24) hypocaloric diet (*n* = 27)	12 weeks	BAC↑ BW BMI Total body fat Visceral adipose tissue Serum glucose levels Serum hepatic growth factor levels *IL-1β* C-reactive protein *Clostridium↑* *Methanosphaera*↑ *Candidatus Soleaferrea*↑	BAC levels are negatively correlated with BW.	[32]
Assessment of the efficacy of probiotic and synbiotic supplementation on intestinal microbiota modulation in adults with prediabetes.	Patients with prediabetes 120/85	Probiotic group (*n* = 27) Synbiotic group (*n* = 29) Placebo group (*n* = 28)	6 weeks	Probiotics: *B. fragilis/E. coli*↑ *Firmicutes/Bacteroidetes*↓	BAC levels are positively correlated with the levels of other beneficial bacteria.	[33]
Assessment of a low-fat vegan diet’s influence on gut microbiota and its correlation with weight and insulin resistance in an overweight cohort.	Participants Age: 25~75 years old BMI: 28~40 kg/m^2^ 168/115	Vegan group (*n* = 84) Control group (*n* = 84)	16 weeks	Vegan group: *B. fragilis*↓ BW Fat mass Visceral fat PREDIM *Faecalibacterium prausnitzii*	BAC levels are negatively correlated with BW.	[34]
Assessment of the influence of a calorie-restricted Mediterranean diet on the gut microbial composition in individuals who are overweight and obese.	Patients with obesity 23/23	NW OB	3 weeks	*B. cellulosilyticus*↑ *B. uniformis*↑ BW Waist circumference Body mass index Fat mass Daily caloric intake *Prevotella stercorea*↑	BAC levels are negatively correlated with BW.	[35]
Investigation of comparative gut microbiota profiles in individuals with dyslipidemia who were overweight and obese: Responses to orlistat and ezetimibe interventions.	174 Volunteers (96 volunteers with dyslipidemia who were overweight and obese) 174/116	Control group (*n* = 31) Patient group (*n* = 27) Orlistat group (*n* = 32) Ezetimibe group (*n* = 26)	12 weeks	BAC↑ Waist circumference TG FBG *Actinomyces*↑	BAC levels are negatively correlated with waist circumference.	[36]
Evaluation of the prebiotic impact of omega-3 supplementation on the gut microbiome.	Patients with obesity 69/69	20 g of inulin fiber 500 mg of omega-3 supplements daily	6 weeks	BAC↑ Butyric acid Iso-butyric acid Iso-valeric acid Total fatty acids Total omega-3/total fatty acids *Coprococcus* spp.↑	BAC levels are positively correlated with increased levels of SCFAs.	[37]
Investigation of predictive alterations in gut microbiota post-short-term low-carbohydrate dietary intervention in patients with obesity.	Participants who were overweight or obese 51/51	ND group (*n* = 25) LCD group (*n* = 26)	12 weeks	BAC↑ BW *Ruminococcaceae Oscillospira*↑ *Odoribacteraceae Butyricimonas*↑ *Porphyromonadaceae Parabacteroides*↑	BAC levels are negatively correlated with BW.	[38]
Investigation of the combined effects of a Medical Food Therapy plant-based diet and intermittent energy restriction on glycemic parameters.	T2D patients without obesity 39/39	20 CMNT group (received a low-calorie human CMNT diet for five consecutive days, 10 days of habitual eating per cycle for 90 days) 19 Control group (continued normal diet)	8 weeks	BAC↑ Hemoglobin A1C BMI FBG *Postprandial* blood glucose levels Medication usage *Parabacteroides*↑ *Roseburia*↑	BAC levels are negatively correlated with BMI.	[39]
Comparative analysis of the effects of traditional vs. commercial kochujang on obesity metrics in adults who are overweight.	Participants 62/48	High-dose Kochujang (*n* = 19) Low-dose Kochujang (*n* = 18) Commercial Kochujang (CK; *n* = 17)	6 weeks	High-dose Kochujang and Low-dose Kochujang groups BAC↑ TC Low-density lipoprotein cholesterol High-density lipoprotein cholesterol Triglyceride levels Waist circumference *Lactobacillus* spp.↑ *Bifidobacterium* spp.↑ *Lactococcus lactis*↑ *Enterococcus faecium*↑	BAC levels are positively correlated with positive health knots, such as reduced waist circumference.	[40]
Evaluation of the effect of *Lactobacillus* salivarius *Ls*-33 on the fecal microbiota composition in adolescents with obesity.	Adolescents with obesity 51/50	1 × 10^10^ CFU L. salivarius *Ls*-33ATCCSD5208 (*n* = 27) Placebo: maltodextrin (*n* = 23)	12 weeks	*B. fragilis*↑ SCFAs↑ *E. rectale*↓	BAC levels were positively correlated with the increase in SCFA levels.	[41]
Detailed analysis of propionate-induced alterations in glucose homeostasis, gut microbiota, plasma metabolome, and immune responses.	Non-diabetic adults who were overweight and obese 14/12	Cellulose Inulin IPE	42 days ×3	Inulin and IPE: *B. uniformis* ↑ *B. caccae* ↑ *B. xylanisolvens* ↑ Insulin resistance IgG *IL-8*	BAC levels are positively correlated with insulin sensitivity and inversely correlated with inflammatory indicators.	[42]
Microbiota shifts in individuals with obesity post very low-energy dietary intervention.	Finnish participants with obesity 16/16	Participants with obesity (BMI > 30 kg/m^2^; six men and ten women)	12 weeks	BAC↑	BAC levels positively correlate with health indicators.	[43]
Examination of the interplay between Korean red ginseng’s effects on metabolic syndrome and gut microbiota alterations.	Patients with metabolic syndrome 60/50	KRG group (*n* = 25) placebo group (*n* = 25)	8 weeks	BAC↑ BMI *Prevotella*↑ *Ruminococcaceae*↑ *Sutterella*↑ *Odobacter*↑	BAC levels are negatively correlated with the BMI and insulin concentrations.	[44]
Assessment of the impact of intermittent fasting on the composition of gut microbiome.	Health volunteers 9/9	Female (*n* = 7) Male (*n* = 2)		*B. fragilis* ↑ *A. muciniphila*↑ Serum fasting glucose levels TC	BAC levels are positively correlated with other beneficial bacteria levels.	[45]

The results presented in the table demonstrate statistical significance. IPE: Inulin-propionate ester; TC: Total cholesterol; BMI: Body Mass Index; T2D: Type 2 diabetes; PREDIM: Predicted clamp-derived insulin sensitivity index from a standard meal test; *IL-1β*: Interleukin 1 beta; FBG: Fasting blood glucose; LBP: Lipopolysaccharide-binding protein; *IL-8*: Interleukin-8; IgG: Immunoglobulin G; SCFAs: Short-chain fatty acids; BW: Body weight. An upward arrow (↑) signifies an increase in the abundance of the specified gut microbe, while a downward arrow (↓) indicates a decrease in the abundance of the specified gut microbe.

**Table 2 ijms-25-00438-t002:** Indirect evidence that *BAC* do not have an anti-obesity effect.

Research Purpose	Numbers of Patients Randomized/Analyzed	Intervention Type (Regimen)	Treatment Period	Main Outcome Measure	Main Conclusion	Reference
To investigate the effects of prebiotic supplementation on body composition, inflammatory markers, bile acids in stool samples, and gut microbiota composition in overweight or children with obesity.	Children who were overweight or obese 42/42	Prebiotic group (*n* = 22) Placebo (*n* = 20)	16 weeks	*B. vulgatus*↓ BW BMI Percent body fat Percent trunk fat *Bifidobacterium*↑ *IL-6*	BAC levels are negatively correlated with the beneficial effects of prebiotic supplementation.	[46]
To investigate the impact of dietary inulin-type fructans (ITF prebiotics) on host metabolism through modulation of the gut microbiota in women with obesity.	Females with obesity 30/30	Inulin/oligofructose 50/50 mix; (*n* = 15) placebo (maltodextrin; *n* = 15)	12 weeks	*B. intestinalis*↓ *B. vulgatus*↓ *Bifidobacterium*↑ *Faecalibacterium* *Rausnitzii*↑ *Propionibacterium*↓ Serum lipopolysaccharide levels	BAC levels are inversely correlated with metabolic improvement due to ITF probiotics.	[47]

BMI: Body mass index; *IL-6*: Interleukin 6. An upward arrow (↑) signifies an increase in the abundance of the specified gut microbe, while a downward arrow (↓) indicates a decrease in the abundance of the specified gut microbe.

### 3.3. Comparison of Gut Microbiota between Individuals with Obesity and Healthy Individuals Demonstrates an Association between BAC and Obesity

To further demonstrate the relationship between BAC and obesity, we included nine clinical trials that analyzed the differences in gut microbiota between populations with and without obesity. Seven of these were studies on adults; two were on children and adolescents; seven studies, while not directly stating their significance, reported lower BAC abundance and diversity in obese populations than in non-obese populations, and three studies directly indicated significance.

The results of a survey of the gut microbiota of 46 pairs of twins and their mothers were revealed. The obese gut microbiota was significantly different (*p* = 0.003) and less diverse than the lean core gut microbiome [48]. In a study by Kasai et al., (2015), a comparison of gut microbiota composition between individuals with and without obesity in a Japanese population showed that *B. faecichinchillae* and *B. thetaiotaomicron* were virtually undetectable in the feces of individuals with obesity and that the proportion of these BAC was significantly higher in individuals without obesity [49]. Additional details are provided in Table 3.

**Table 3 ijms-25-00438-t003:** Comparison of gut microbiota between individuals with obesity and healthy individuals demonstrating the indirect metabolism-modulation function of BAC.

Research Purpose	Main Outcome Measure	Results (Relationship with BAC)	Main Conclusion	Reference
Objective: To characterize the gut microbiota and short-chain fatty acids (SCFAs) in participants with obesity from Xinjiang, China. Summary: This study highlights regional variations in gut microbial composition related to obesity.	16S rRNA sequencing and microbial diversity and community analyses 68 individuals with obesity 31 controls	Individuals with obesity↑ *B*. *fragilis*↓ *Gemmiger*↑ *Dialister*↑ *Megamonas*↑ *Anaerostipe*↑ *Blautia*↑ *Gemmiger*↓ *Bifidobacterium*↓ *Prevotella*↓	The gut microbiota profile of obese patients was characterized by enrichment of *Lactobacillus* and the reduction in the diversity and the depletion of *Actinobacteria,* BAC, *Bifidobacterium*, and *B. fragilis*.	[36]
Objective: To identify core gut microbiota in obese and lean twins and their association with obesity. Summary: This study investigated twins discordant in obesity to reveal key microbial players associated with obesity.	16S rRNA gene sequencing 31 monozygotic twin pairs 23 dizygotic twin pairs and their mothers (*n* = 46)	Participants with obesity: BAC↓ *Actinobacteria*↑ Carbohydrate, lipid, and amino acid metabolism↓	The deviations from a core microbiome at function level are associated with different physiological states.	[48]
FObjective: To determine the relationship between gut microbiota composition and obesity in a Japanese population. Summary: This study explores unique microbial patterns associated with obesity in the Japanese demographic.	T-RFLP analysis and 16S rRNA sequencing 23 participants without obesity BMI < 20 kg/m^2^ 33 participants with obesity BMI ≥ 25 kg/m^2^	Participants without obesity: *B. faecichinchillae*↑ *B. thetaiotaomicron*↑ *Blautia wexlerae*↑ *Clostridium bolteae*↑ *Flavonifractor plautii*↑	Gut microbial properties differ between obese and non-obese subjects, suggesting that gut microbiota composition is related to obesity.	[49]
Objective: To elucidate the gut microbial composition in normal adolescents and those with obesity. Summary: This research uncovered microbial signatures specific to obesity during adolescence.	16S rRNA gene sequencing 67 adolescents with obesity (BMI ≥ 30 kg/m^2^ or ≥ 99th BMI percentile) 67 normal adolescents (BMI < 25 kg/m^2^ or <85th BMI percentile)	Adolescents with obesity BAC↓ Prevotella↓ TG↑ TC↑ hs-CRP↑	A significant association between the composition of several bacterial taxa and childhood obesity was revealed.	[50]
Objective: To understand the mechanisms of obesity development by comparing the gut microbiota of children with obesity with that of healthy controls. Summary: This research highlights distinct gut microbial features associated with obesity.	16S rRNA gene sequencing, enterotypes Children with obesity aged 3 to 18 years (*n* = 87) healthy children aged 3 to 18 years (*n* = 56)	Children with obesity: BAC↓ *Firmicutes/Bacteroidetes* ↑ *Enterococcus*↑ *Blautia*↑ *Sutterella*↑ *Klebsiella*↑ *Parabacteroides*↓ *Anaerotruncus*↓ *Coprobacillus*↓	BAC are associated with weight loss, and are recommended as a dietary prebiotic and probiotic supplement as an adjunct to obesity treatment.	[51]
Objective: To identify microbial and metabolic factors associated with obesity through MWAS and serum metabolomics profiling. Summary: This comprehensive approach uncovers microbial and metabolic contributors to obesity in young Chinese individuals.	shotgun sequencing 72 individuals with obesity (body mass index (BMI), 36.78 ± 4.46 kg/m^2^; age, 23.6 ± 3.7 years) 79 controls (BMI, 20.2 ± 1.3 kg/m^2^; age, 23.2 ± 1.8 years)	Controls *B*. *thetaiotaomicron*↑ *B*. *uniformis*↑ *B. xylanisolvens*↑ *B*. *ovatus* ↑ *Akkermansia muciniphila*↑ *Fecalibacterium prausnitzii*↑	The abundance and diversity of BAC were significantly lower in obese individuals than in normal individuals.	[52]
Objective: To study gut microbiota alterations specific to childhood obesity. Summary: This research reveals microbial markers and shifts associated with obesity during childhood.	16S rRNA gene sequencing Children with obesity aged 6 to 16 years (*n* = 42) Children of normal weight (*n* = 36)	Children with obesity: *B. vulgatus*↓ *B. stercoris*↓ *Firmicutes/Bacteroidetes*↑ (*p* < 0.0001)	Reduced numbers of BAC in the gut may be linked to childhood obesity.	[53]
Objective: To identify unique gut microbiota characteristics in patients who were overweight/obese and compare them with those in normal weight controls in Sardinia. Summary: This research uncovers microbial signatures specific to obesity in this geographical region.	16S rRNA gene sequencing Patients who were overweight/obese (*n* = 46) normal weight participants (*n* = 46)	Patients who were overweight/obese: BAC↓ *Flavobacteriaceae*↓ *Porphyromonadaceae*↓ *Sphingobacteriaceae*↓ *Flavobacterium*↓ *Rikenella* spp.↓ *Pedobacter* spp.↓ *Parabacteroides* spp.↓	Body fatness and waist circumference were negatively correlated with BAC.	[54]
Objective: To explore interactions between the gut microbiome, metabolites, and brain network metrics. Summary: This investigation sheds light on how the gut microbiota and its metabolic products influence brain function, potentially linking gut–brain interactions to obesity.	16S rRNA gene sequencing 287 participants with obesity and without obesity (male *n* = 99, female *n* = 198)	Participants with obesity: BAC↓ Microbial diversity↓ *Prevotella*/BAC↑ Fecal tryptophan↓	Observed significant taxonomic changes at the genus level associated with obesity. BAC↓ (*p* < 0.001)	[55]

The results presented in the table demonstrate statistical significance. BMI: Body mass index; hs-CRP: High-sensitivity C-reactive protein; SCFAs: Short-chain fatty acids; TC: Total cholesterol; TG: Total triglyceride; T-RFLP: Terminal restriction fragment length polymorphism. An upward arrow (↑) signifies an increase in the abundance of the specified gut microbe, while a downward arrow (↓) indicates a decrease in the abundance of the specified gut microbe.

### 3.4. In Vivo Direct Evidence of the Anti-Obesity Effects of BAC

Animal experiments published in 2021 had a high frequency of publications; five were conducted on C57BL/6J mice, and only one on rats; >90% of the studies used male animals. The duration of the intervention ranged from 7 to 18 weeks. Consistent evidence from various animal studies indicates that certain BAC strains can significantly reduce key obesity-related metrics, including body weight, fat accumulation, and adverse serum profiles. 

The six preclinical studies involved *B. uniformis* CECT 7771, *B. uniformis* CBA7346, *B. vulgatus* SNUG 40005, *B. acidifaciens* JCM10556, *B. dorei* DSM17855, and *B. vulgatus* ATCC8482. Cano et al., and Agusti et al., compared the ability of different BAC strains to produce cytokines. Notably, administration of *B. uniformis* CECT 7771 was found to significantly reduce body weight gain in HFD-fed mice (*p* < 0.05) and caloric intake during the binge phase (*p* < 0.001) [56,57]. In another study, administration of *B. vulgatus* SNUG 40,005 for 18 weeks significantly reduced the body weights of obese mice (*p* = 0.004) [58]. In the study by Yang et al., (2017), oral administration of *B. acidifaciens* resulted in a significant reduction in body weight (*p* < 0.001) and adiposity (*p* < 0.05) in Atg7 ^ΔCD11c^ mice [14]. Administration of *B. uniformis* CBA7346 significantly reduced body and liver weights (*p* < 0.05) [59]. Yoshida et al., (2021) found that combined gavage of *B. dorei* DSM17855 and *B. vulgatus* ATCC8482 significantly suppressed obesity and weight gain [60]. Further details are listed in Table 4.

**Table 4 ijms-25-00438-t004:** In vivo evidence of anti-obesity effect of BAC.

*Bacteroides* Strain	Source	Subject	Intervention Type (Regimen)	Treatment Period	Main Outcome Measure	Main Conclusion	Reference
*B. acidifaciens* JCM10556	Human feces	C57BL/6, CD11c-Cre, Villine-Cre, Atg7f/f and LysM-Cre mice	HFD + PBS HFD + 5 × 10^9^ cfu *B. acidifaciens* JCM10556	10 weeks	BW and fat mass Insulin Bile acid-TGR5-PPARα axis Dipeptidyl peptidase-4 Glucagon-like peptide-1	*B. acidifaciens* JCM10556 helps prevent metabolic disorders such as diabetes and obesity.	[14]
*B. uniformis* CECT7771	Infant stool	Male C57BL/6J mice	Nor HFD Nor + 5.0 × 10^8^ cfu *B. uniformis* CECT 7771 HFD + 5.0 × 10^8^ cfu *B. uniformis* CECT 7771	7 weeks	BW Adipose tissue weight Hepatic steatosis Macrophage functionality Dendritic cell functionality	*B. uniformis* CECT7771 improves metabolic and immune dysfunction associated with intestinal ecological dysregulation.	[56]
*B. uniformis* CECT7771	Infant stool	Adult male Wistar Kyoto rats (170–200 g)	SD IF IF + 1 × 10^8^ cfu *B. uniformis* CECT 7771	18 days	Caloric intake Anxiety-like behavior Dopamine	*B. uniformis* CECT 7771 helps control compulsive eating by affecting brain reward systems.	[57]
*B. uniformis* CBA7346	Human feces	Male C57BL/6J mice	Nor + PBS Nor + 1 × 10^6^ cfu *B. uniformis* HFD + PBS HFD + 1 × 10^6^ cfu *B. uniformis*	12 weeks	Body and liver weights Blood lipids Liver injury and steatosis Hepatic lipid metabolism	*B. uniformis* CBA7346 mitigates HFD-induced NAFLD, thereby modulating LPS release, lipid-related proteins, and insulin sensitivity.	[59]
*B. dorei* DSM17855 and *B. vulgatus* ATCC8482	Human feces	Male C57BL/6J mice	Nor HFD HFD + 2.5 × 10^9^ cfu *B. dorei* + *B. vulgatus* mix	12 weeks	BW Blood glucose BAT weight UCP1 Macrophage number	*B. dorei* DSM17855 and *B. vulgatus* ATCC8482 enhance branched-chain amino acid catabolism in brown adipose tissue	[60]

The results presented in the table demonstrate statistical significance. UCP1: Uncoupling protein 1; BAT: Brown adipose tissue; BW: Body weight.

### 3.5. Studies on Direct Anti-Inflammatory Therapy with BAC

Recent studies have shown that white adipose tissue in patients with obesity develops low-grade chronic inflammation, which induces insulin resistance [61]. Due to the close relationship between obesity and inflammation, we included six preclinical studies in which BAC were directly gavaged. All experiments were conducted with mice; the minimum duration of the experiment was five days, and the maximum was 21 days. Only three experiments clearly labeled the gavaged BAC concentration, which ranged from 1 × 10^9^ CFU/mL to 7 × 10^9^ CFU/mL. In these studies, BAC significantly upregulated the expression of anti-inflammatory cytokines and downregulated the expression of pro-inflammatory cytokines. These studies suggest that BAC have therapeutic potential for preventing intestinal inflammatory diseases in animal models (Table 5).

**Table 5 ijms-25-00438-t005:** Study of direct anti-inflammation treatments with BAC.

*Bacteroides* Strain	Source	Subject	Intervention Type (Regimen)	Treatment Period	Main Outcome Measure	Main Conclusion	Reference
*B. fragilis* NCTC9343	Human feces	Male C57BL/6J mice	DSS DSS + 5 × 10^9^~7 × 10^9^ CFU *B. fragilis* NCTC9343	8 days	*IL-1β* CCR5 Number of colonic tumors TLR2 signaling	This study highlights *B. fragilis*’ protective role against weight loss, colonic histopathological changes, and inflammation in colitis-associated colon cancer.	[62]
*B. thetaiotaomicron* DSM 2079	Human feces	Female and male C57BL/6J mice	Control DSS DSS + 3 × 10^10^ CFU *B. thetaiotaomicron* DSM 2079	8–14 days	inflammation BW Colon length	This study demonstrates that *B. ovatus* monotherapy is a consistent and effective approach for colitis management, surpassing traditional FMT.	[63]
*B. fragilis* (HCK-B3, ATCC25285) and *B. ovatus* (ELH-B2, JCM5824)	Human feces	Female C57BL/6J mice	1 × 10^9^ CFU *B. fragilis* HCK 1 × 10^9^ CFU *B. fragilis* 25285 1 × 10^9^ CFU *B. ovatus* ELH 1 × 10^9^ CFU *B. ovatus* JCM5824	5 days	*TNF* *IL-10* FITC-Dextran NF-κB	This research underscores the promise of BAC as a therapeutic agent for intestinal inflammatory diseases.	[64]
*B. ovatus* ATCC 8483	Human feces	Male C57BL/6J mice	*B. ovatus* *B. thetaiotaomicron* *B. vulgatus* *B. ovatus* + *B. thetaiotaomicron* + *B. vulgatus* mix	9 days	Increased proliferation of epithelial cells Hyperplastic crypts inflammation	This study demonstrates that *B. ovatus* monotherapy is a consistent and effective approach for colitis management, surpassing traditional FMT.	[65]
*B. fragilis* NCTC9343	Human feces	WT (129S6/SvEvTac) and Rag2−/− (129S6/SvEvTac-Rag2tm1Fwa) mice	WT + *B. fragilis* WT + PSA Rag + PSA WT + PBS Rag + PBS	21 days	T-cell populations *IL-10* *IFNγ*	This research underscores the protective role of *B. fragilis* NCTC9343 against nervous system inflammatory diseases.	[66]
*B. vulgatus* FJS7K1	Human feces	Male C57BL/6J mice	Control LPS LPS + *B. vulgatus* 5K1 LPS + *B. vulgatus* 7K1 LPS + *B. vulgatus* 11B4 LPS + *B. vulgatus* 51K1	5 days	Number of Treg cells Intestinal epithelial integrity *IL-6* *IL-10* *TNF* SCFAs	This study highlights *B. vulgatus* FTJS7K1 as a potential agent for mitigating acute inflammation and intestinal injury through microbial community modulation and cytokine regulation.	[67]

The results presented in the table demonstrate statistical significance. SCFAs: Short-chain fatty acids; *IL-6*: Interleukin-6; *TNF*: Tumor necrosis factor α; *IL-10*: Interleukin-10; *IFNγ*: Interferon-γ; FITC-Dextran: Fluorescein isothiocyanate; NF-κB: Nuclear factor κB; TLR2: Toll-like receptor 2; CCR5: C-C chemokine receptor 5; LPS: Lipopolysaccharide; PSA: Polysaccharide A; WT: Wild type.

### 3.6. Quality Assessment and Scores

After a quality assessment of the selected studies using the modified Downs and Black checklist, only 9 of the 17 investigational trials on humans scored a superior grade [28,29,31,32,33,34,38,40,44]. Six studies were assessed as high quality [30,35,39,41,42,43]. Eight experiments provided sufficient details on withdrawal and exit [28,31,32,33,34,38,40,44]. We also considered participant empowerment, and all experiments mentioned having gone through an ethical approval process and obtained informed consent from the participants (Table 6). We present our quality assessment for all included studies. This assessment highlights that all selected articles achieved 100% compliance with reporting standards, and 50% of these studies met the criteria for adequate statistical power (Figure 2).

**Table 6 ijms-25-00438-t006:** Quality assessment and scores of included studies using Downs and Black quality checklist.

		Larsen et al. [41]	Simoes et al. [43]	Han et al. [28]	Song et al. [29]	Gonzalez-Sarrias et al. [30]	Chambers et al. [42]	Ejtahed et al. [31]	Henning et al. [32]	Ozkul et al. [45]	Kassaian et al. [33]	Kahleova et al. [34]	Pisanu et al. [35]	Seong et al. [44]	Vijay et al. [37]	Zhang et al. [38]	Luo et al. [39]	Han et al. [40]
Q1	Hypothesis/aim/objective clearly described	1	1	1	1	1	1	1	1	1	1	1	1	1	1	1	1	1
Q2	Main outcomes in Introduction or Methods	1	1	1	1	1	1	1	1	1	1	1	1	1	1	1	1	1
Q3	Patient characteristics clearly described	1	1	1	1	1	1	1	1	1	1	1	1	1	1	1	1	1
Q4	Interventions of interest clearly described	1	1	1	1	1	1	1	1	1	1	1	1	1	1	1	1	1
Q5	Principal confounders clearly described	1	1	1	1	1	1	1	1	1	1	1	1	1	1	1	1	1
Q6	Main findings clearly described	1	1	1	1	1	1	1	1	1	1	1	1	1	1	1	1	1
Q7	Estimates of random variability provided for main outcomes	1	1	1	1	1	1	1	1	-	1	1	1	1	1	1	1	1
Q8	All adverse events of intervention reported	0	0	1	1	1	1	1	1	0	1	1	1	1	0	1	1	1
Q9	Characteristics of patients lost to follow-up described	0	-	1	-	0	-	1	1	-	1	1	-	1	0	1	0	1
Q10	Probability values reported for main outcomes	1	1	1	1	1	1	1	1	1	1	1	1	1	1	1	1	1
Q11	Subjects Study participants asked to participate were representative of the source population	1	1	1	1	1	1	1	1	1	1	1	1	1	1	1	1	1
Q12	Subjects Study participants prepared to participate were representative of the source population	1	1	1	1	1	1	1	1	1	1	1	1	1	1	1	1	1
Q13	Location and delivery of study treatment were representative of the source population	1	1	1	1	1	1	1	1	1	1	1	1	1	1	1	1	1
Q14	Study participants blinded to treatment	1	-	-	1	1	1	1	1	-	1	1	-	1	0	1	1	1
Q15	Blinded outcome assessment	1	-	-	1	1	1	1	1	-	1	1	-	1	1	1	1	1
Q16	Any data dredging clearly described	1	1	1	1	1	1	1	1	1	1	1	1	1	1	1	1	1
Q17	Analyses adjusted for differing lengths of follow-up	0	0	1	1	1	1	1	1	1	1	1	1	1	1	1	0	1
Q18	Appropriate statistical tests performed	1	1	1	1	1	1	1	1	1	1	1	1	1	1	1	1	1
Q19	Compliance with interventions was reliable	1	1	1	1	1	1	1	1	1	1	1	1	1	1	1	1	1
Q20	Outcome measures were reliable and valid	1	1	1	1	1	1	1	1	1	1	1	1	1	1	1	1	1
Q21	All participants recruited from the same source population	1	1	1	1	1	1	1	1	1	1	1	1	1	1	1	1	1
Q22	All participants recruited over the same time period	1	1	1	1	1	1	1	1	1	1	1	1	1	1	1	1	1
Q23	Participants randomized to treatment(s)	1	-	1	1	1	1	1	1	-	1	1	1	1	-	1	1	1
Q24	Allocation of treatment concealed from investigators and participants	1	-	1	1	1	1	1	1	-	1	1	0	1	-	1	0	1
Q25	Adequate adjustment for confounding	1	0	1	1	1	1	1	1	1	1	1	0	1	1	1	1	1
Q26	Losses to follow-up taken into account	0	0	1	1	0	0	1	1	0	1	1	1	1	0	1	1	1
Q27	Sufficient power to detect treatment effect at a significance level of 0.05	0	0	1	0	0	0	0	1	0	1	1	1	1	1	0	1	1
TOTAL		22	20	26	25	24	24	26	27	19	27	26	23	26	19	26	24	27

The ‘-’ symbol indicates that this clinical laboratory course can be used without analyzing this question. Count one point.

**Figure 2 ijms-25-00438-f002:**
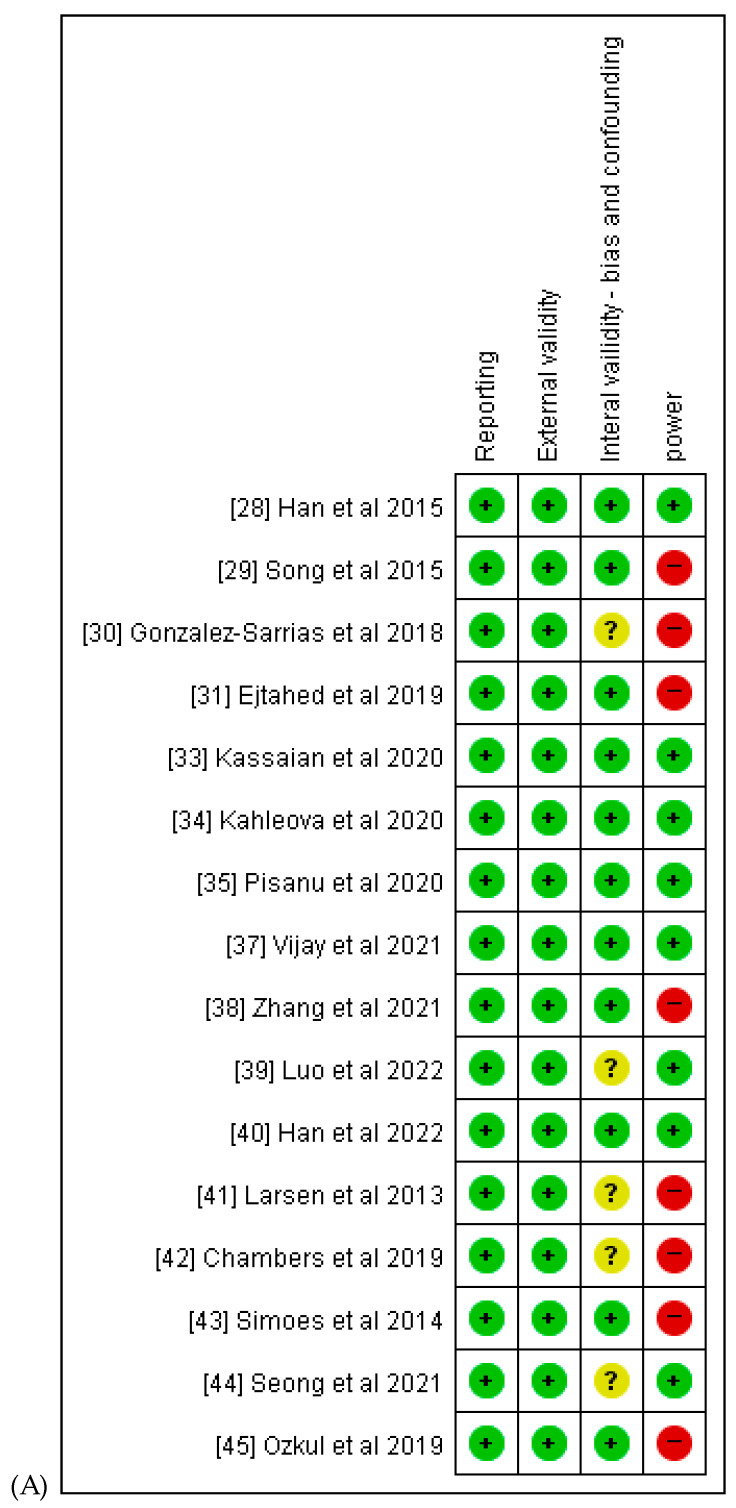
Quality assessment. (**A**) presents quality assessment summary: review authors’ judgments about each risk of bias item for each included studies low risk (+: green), high risk (-: red), or unclear risk (?: yellow); (**B**) indicates quality assessment graph: review authors’ judgments about each assessment item presented as percentages across all included studies.

## 4. Discussion

*Bacteroides* spp. (BAC) constitute a significant proportion of the human gut microbiota and play a pivotal role in maintaining host metabolic homeostasis and immune regulation. These microbes aid in lipid metabolism, potentially influencing obesity outcomes and systemic inflammation [68]. This review represents the first thorough examination of BAC in obesity, detailing their correlation with body weight and prevalence in diverse populations across multiple countries. It also includes an analysis of preclinical studies on obesity and inflammation, shedding light on the potential mechanisms of action.

There is a recent study demonstrating the ability of BAC to degrade a wide range of polysaccharides and summarizing the utilization of BAC and their beneficial effects on intestinal health that supports our review [69]. This study demonstrated the BAC catabolism of polysaccharides to oligosaccharides or short-chain fatty acids (SCFAs). Fermentation of oligosaccharides and monosaccharides by SCFA-producing bacteria increases the luminal concentrations of acetate, butyrate, and propionate, which are beneficial effects of SCFAs on intestinal inflammation and immunity. 

Furthermore, our review summarized clinical studies and randomized controlled trials of human anti-obesity treatments and focused on differences and relationships between body weight and BAC. In our review, all included preclinical studies presented positive improved results. In 70% of the clinical trials, weight loss in humans after treatment was accompanied by increased BAC richness. However, some studies have shown opposite results, of which two examples are provided in this review. We speculate that BAC richness is highly susceptible to changes caused by other drugs or foods, and we suspect an inhibitory relationship between probiotics and BAC. However, we still need to shed light on the actual evidence for this conjecture, and more experimental evidence is required. In addition, a study on microbial enterotypes reported that enterotypes may influence an individual’s ability to lose weight when following a specific diet [6]. We previously speculated that the difference in each individual characteristic affects the amount of BAC in the body. This study suggests that stratifying individuals according to two microbial enterotypes (dominance of either Prevotella or BAC) may help predict the response to diet and medication. It is a challenging point that individual differences can influence our judgment and conclusions regarding the results. Although preclinical studies have predominantly indicated a consistent anti-obesity effect of BAC, human trials present more heterogeneous outcomes. There are also studies that show that human dietary preferences in animal husbandry practices may influence domesticated animals [70]. Dietary homogenization among different species may lead to convergence of microbial characteristics. This discrepancy underscores the complexity of translating microbial interventions from animal models to humans, influenced by factors such as individual microbiome composition, diet, and lifestyle.

Because there are no sufficient clinical trials used to demonstrate the anti-obesity effect of BAC, the preclinical studies were reviewed to overcome this lack of studies. After a review of the preclinical studies focused on the mechanism of BAC in obesity was conducted, we concluded that BAC significantly prevents obesity in the animal model, particularly in mice. Our analysis underscores the potential of certain BAC, such as *B. uniformis* and *B. vulgatus*, in mitigating obesity-related parameters. For example, *B. uniformis* has been shown to reduce lipid accumulation and modulate adipose tissue inflammation, whereas *B. vulgatus* appears to influence body weight regulation, possibly through interactions with gut mucin dynamics. In our analyses, the BAC measure had to be at least 1 × 10^6^ CFU/day and lasted at least 7 weeks to present validity in the body weight results. We also found that BAC improved immune dysfunction, promoted the growth of *Akkermansia* by utilizing metabolites from mucin degradation, increased serum insulin levels, and prevented obesity by promoting the catabolism of circulating branched-chain amino acids in the brown adipose tissue. Notably, Agustí et al., administered *B. uniformis* CECT 7771 to rats for 18 days and found that it affected the brain’s reward response, improved overeating, and reduced anxiety behavior. These effects are produced, at least in part, by altering the levels of dopamine, 5-hydroxytryptamine, and norepinephrine in the nucleus ambiguous, as well as the expression of dopamine D1 and D2 receptors in the prefrontal cortex and gut [57]. Furthermore, certain BAC can modulate systemic and local inflammation. *B. vulgatus* FTJS7K1 have been observed to downregulate pro-inflammatory cytokines and potentially contribute to maintaining gut barrier integrity, thereby potentially reducing low-grade systemic inflammation often observed in obesity scenarios [67]. The mechanisms underlying these effects are not entirely clear and may involve complex interactions between gut microbiota, host metabolism, and systemic inflammation. Further research is needed to clarify these mechanisms and determine whether similar effects occur in humans.

This study had some limitations. First, this review was constrained by the scarcity of long-term human trials and inconsistencies in the study design, particularly concerning probiotic strains, dosages, and participant demographics. These variations complicate the interpretation of the therapeutic potential of BAC in obesity management. Second, attempts were made to identify all randomized controlled trials on the increase in BAC following treatment efficacy, but potential incompleteness of citation tracking may have resulted in some relevant randomized controlled studies being overlooked. In addition, despite clear eligibility criteria, these studies need to be more consistent regarding this study’s design, probiotic strain, form and duration of application, sample population, and outcomes. Data are often not reported because many studies have not reported positive results for BAC after pharmacological interventions. 

The quality assessment of the current studies indicated variability in the quality of the available research. While many studies were conducted, some lacked sufficient rigor, highlighting the need for more high-quality, consistent, and replicable research in this field. Most studies have been short-term, ranging from 4 to 24 weeks. Considering that obesity-related inflammation is a chronic disease, long-term evaluation of treatment is recommended. An apparent strength of this study is that our search included combinations of obesity and BAC and combinations of BAC alone or combinations where it was determined that BAC produced an effect, if available.

This review provides direct and indirect evidence of the effectiveness of BAC supplementation in treating obesity. In conclusion, BAC offers a promising avenue for treating obesity and inflammation, with certain BAC strains having specific therapeutic potential. However, the existing body of research is characterized by variability, necessitating more comprehensive, standardized, and long-term clinical trials. Future investigations are needed to elucidate the mechanistic pathways underlying the anti-obesity effects of BAC, assess the feasibility and acceptance of interventions, such as fecal microbiota transplant (FMT), and establish protocols for overcoming challenges in post-administration microbial colonization. Through this, we can progress toward targeted obesity therapies that harness the beneficial capacities of the gut microbiota.

## 5. Conclusions

This review rigorously evaluated the effect of BAC supplementation on obesity-related markers, revealing a nuanced relationship between specific species and strains. The analysis delineates the effects of several BAC strains, notably *B. uniformis* (CECT 7771 and CBA 7346), *B. fragilis* (HCK-B3, ATCC25285, and NCTC9343), *B. vulgatus* (SNUG 40005, ATCC 8482, and FTJS7K1), *B. ovatus* (ELH- B2, JCM5824, and ATCC 8483), *B. acidifaciens* JCM 10556, *B. dorei* DSM 17855, and *B. thetaiotaomicron* DSM 2079, on the modulation of BW, waist circumference (WC), adiposity, and systemic levels of pro-inflammatory cytokines.

Intriguingly, these anti-obesity properties appear to be species- and strain-specific, underscoring the biological complexity and the necessity for precision when considering probiotic interventions. Despite promising indications from preclinical studies, there is a lack of human clinical trials that directly examine the effects of BAC supplementation on obesity. This gap highlights an unresolved dichotomy in the field, wherein empirical support is burgeoning, yet a definitive clinical consensus remains elusive.

Therefore, the therapeutic potential of BAC in managing obesity requires further targeted longitudinal human studies. Future research should prioritize identifying and validating the most efficient BAC strains for obesity mitigation and investigate their mechanistic pathways, optimal dosages, and administration protocols. Moreover, exploring synergistic or antagonistic interactions between different BAC strains and broader microbiome-host dynamics will be instrumental in harnessing the full therapeutic arsenal of these microbes.

## Figures and Tables

**Figure 1 ijms-25-00438-f001:**
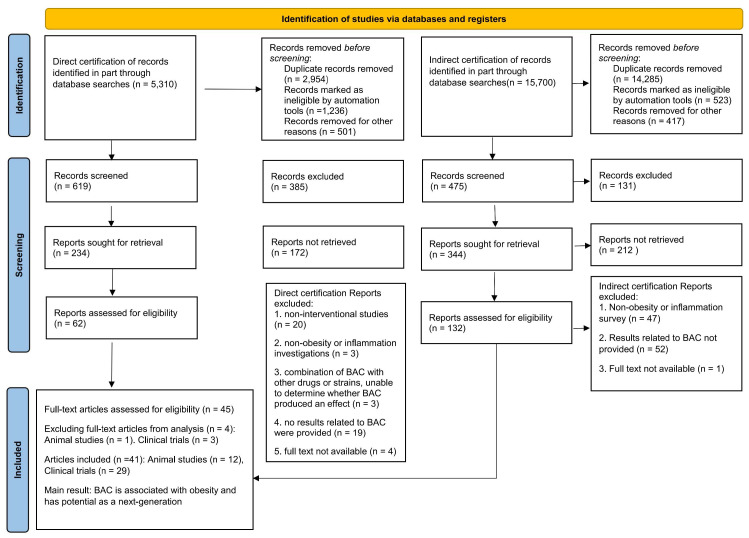
Flowchart of the process for screening articles on systematic reviews that directly and indirectly demonstrate the effects of *Bacteroides* spp. on human obesity.

## Data Availability

Data are available upon request to the corresponding author.

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
