# Peer review of "Direct and Indirect Evidence of Effects of Bacteroides spp. on Obesity and Inflammation"

_ijms, 2023, doi:10.3390/ijms25010438_

Round 1
Reviewer 1 Report
Comments and Suggestions for Authors
According to the paper authored by Liangliang Wu and colleagues "A Systematic Review of Effects of Bacteroides spp. on Obesity and Inflammation: Direct and Indirect Evidence". Global public health is facing an increasing number of metabolic disorders. Despite its established significance, the relationship between the gut microbiome and obesity, as well as its exact metabolic functions, remains unclear. In both preclinical and clinical studies, Bacteroides (BAC) species have demonstrated substantial anti-obesity effects; however, their underlying biochemical pathways have not yet been fully elucidated. A systematic review of recent studies is presented in this paper in order to provide an overview of the direct and indirect evidence supporting the beneficial role of Bacillus spp in the treatment of obesity. First, we consolidated existing knowledge about how gut microbiota influences obesity; second, we evaluated the effectiveness of BAC species in combating obesity; and third, we explored the potential mechanisms by which BAC species can affect obesity and lipid metabolism. In this review, BAC species have been reported to ameliorate obesity and inflammation in preclinical studies and detailed methodologies for in vivo experiments are presented. We examined studies documenting shifts in the diversity and concentration of BAC species after successful obesity treatment, validating BAC's role in obesity. Further evidence of their crucial role in obesity is provided by the significant variations in the profiles of BAC species among individuals with different levels of adiposity. In addition, certain BAC-derived metabolites have been implicated in the regulation of pathways related to the host's lipid metabolism. Furthermore, the administration of BAC increases dermal ischemia in response to LPS stimulation, suggesting its usefulness as adjunctive therapy for obesity. I would like to make a few comments regarding the present manuscript.
According to the references, the authors have not followed the guidelines.
Your manuscript should be changed to reflect that this is a systematic review.
Does this systematic review appear in PROSPERO? It is a quality requirement
Tables are huge, and I believe that p_0.05 is of no importance
Why were the results divided into five sections, and what are the main questions the researchers have?
Other programs, such as RevMan, may be used to summarize the quality
In order to evaluate the effects, a metaanalysis report showing the actual effects would be helpful
There is a lack of new information in the discussion, as the main questions related to PICOS are not addressed
Author Response
According to the paper authored by Liangliang Wu and colleagues "A Systematic Review of Effects of Bacteroides spp. on Obesity and Inflammation: Direct and Indirect Evidence". Global public health is facing an increasing number of metabolic disorders. Despite its established significance, the relationship between the gut microbiome and obesity, as well as its exact metabolic functions, remains unclear. In both preclinical and clinical studies, Bacteroides (BAC) species have demonstrated substantial anti-obesity effects; however, their underlying biochemical pathways have not yet been fully elucidated. A systematic review of recent studies is presented in this paper in order to provide an overview of the direct and indirect evidence supporting the beneficial role of Bacillus spp in the treatment of obesity. First, we consolidated existing knowledge about how gut microbiota influences obesity; second, we evaluated the effectiveness of BAC species in combating obesity; and third, we explored the potential mechanisms by which BAC species can affect obesity and lipid metabolism. In this review, BAC species have been reported to ameliorate obesity and inflammation in preclinical studies and detailed methodologies for in vivo experiments are presented. We examined studies documenting shifts in the diversity and concentration of BAC species after successful obesity treatment, validating BAC's role in obesity. Further evidence of their crucial role in obesity is provided by the significant variations in the profiles of BAC species among individuals with different levels of adiposity. In addition, certain BAC-derived metabolites have been implicated in the regulation of pathways related to the host's lipid metabolism. Furthermore, the administration of BAC increases dermal ischemia in response to LPS stimulation, suggesting its usefulness as adjunctive therapy for obesity. I would like to make a few comments regarding the present manuscript.
According to the references, the authors have not followed the guidelines.
> We greatly appreciate your insightful feedback regarding this particular aspect of our manuscript. In response to your valuable suggestion, we have carefully revised the original article to incorporate the suggested change.
Your manuscript should be changed to reflect that this is a systematic review.
> We acknowledge your observation that our manuscript should more clearly reflect its nature as a systematic review. Our review has been carried out systemically, but in technically, it is more in line literature review than in the form of a typical type of systematic review. We are fully aware that the expression “systematic” can cause confusion, and accordingly, we carefully revised and deleted the expression “systematic” or “systematically” throughout the full manuscript to reduce methodological confusion.
Does this systematic review appear in PROSPERO? It is a quality requirement
> We greatly appreciate your insightful and understand your concern. As we know, PROSPERO only accept two types of protocol; a systematic review on human studies or a review of animal studies relevant to human health. Unfortunately, our review was conducted both clinical trials and preclinical studies using animal model. Moreover, literature reviews that use a systematic search, our review, does not meet acceptance criteria for PROSPERO. For this reason, we have applied for registration with PROSPERO only for human research, but it has not been accepted yet.
Tables are huge, and I believe that p_0.05 is of no importance
> We understand your concern about the extensive tables and the inclusion of p_0.05 results. We will streamline the tables for clarity and focus on more significant results, ensuring they succinctly convey the necessary information.
We deleted all p-values in the table and added 'The results presented in the table demonstrate statistical significance.' in the ellipsis after tables.
Why were the results divided into five sections, and what are the main questions the researchers have?
> In light of the known anti-inflammatory effects of Bacteroides spp. (BAC) and the established link between inflammation and obesity, our study aims to explore the potential role of BAC in obesity treatment. Due to the absence of clinical trials directly investigating BAC's efficacy in this domain, we searched and reviewed both clinical trials and preclinical studies. Because of its heterogeneity, we analyzed and presented separately and we have organized our article into five distinct sections to provide a thorough examination of this topic.
- The first section collates studies that have indirectly demonstrated changes in BAC following various treatments.
- Second section organizes a number of examples that are contrary to the results of first section.
- The third section is dedicated to exploring differences in BAC abundance within the gut microbiota of obese individuals compared to healthy controls.
- Given the lack of direct clinical trials, the fourth section reviews preclinical studies suggesting BAC's potential in treating obesity. These studies, predominantly using mouse models, provide valuable insights into BAC's role in weight management, explored through appropriate medical subject headings.
- The firth section reviews preclinical studies where BAC has been found to directly impact inflammation. This segment focuses on understanding the mechanisms through which BAC may influence inflammatory processes.
Through this structured approach, our article aims to comprehensively assess the multifaceted role of BAC in obesity management, considering both its direct and indirect effects. This examination seeks to contribute to the understanding of BAC's potential therapeutic applications in the context of obesity and associated metabolic disorders. To enhance understanding, we combined the first and second section. As a results, the revised structure is followed;
- Clinical trials for indirect evidence: changes in BAC following treatment for obesity.
- Clinical trials: comparison of BAC abundance between obese and healthy human.
- Preclinical trials: direct evidence of BAC for obesity
- Preclinical trials: effect of BAC for inflammation related to obesity
Other programs, such as RevMan, may be used to summarize the quality
> We appreciate your suggestion to use RevMan for summarizing the quality of included studies. We presented the quality of included studies using RevMan. Because there was no limitation of article type to randomized controlled trials, RoB tool by Cochrane handbook was not suitable for the quality assessment. Therefore, modified Down and Black checklist was used for the quality assessment, so that, the figure details were modified to matched to assessment tool.
In order to evaluate the effects, a metaanalysis report showing the actual effects would be helpful
> We agree that a meta-analysis report showing the actual effects. We considered and tried to conduct meta-analysis, unfortunately, the extracted data were not possible to perform meta-analysis. Reasons for not suitable include the use of different outcome measurement for each study or the use of units that cannot be unified, the absence of suitable number to be analyzed, too few corresponding studies, and so on.
There is a lack of new information in the discussion, as the main questions related to PICOS are not addressed
> We recognize your concern regarding the discussion section. To address this, we will revise it to include more novel insights, particularly focusing on addressing the main questions related to PICOS (Population, Intervention, Comparison, Outcomes, and Study Design). This will ensure a more thorough and insightful discussion of our findings.
We are committed to improving our manuscript in line with your valuable feedback and believe these changes will significantly enhance the quality and clarity of our work.
Reviewer 2 Report
Comments and Suggestions for Authors
The paper is well written and well organized. Minor revision is needed, mostly in the introduction.
L13 Bacteroides (BAC) spp.
Bacteroides spp. (BAC)
L16 BAC spp.
Just BAC, remove spp. It needs to be corrected throughout the entire manuscript (L42, L53, L56, L65, L83, L134,166,271, 316…).
L27 TNF-α
TNF is still referred to as TNF-α in numerous newly published scientific papers, almost 2 decades after the cytokine was renamed. Please see Grimstad, Ø., 2016. Tumor necrosis factor and the tenacious α. JAMA dermatology, 152(5), pp.557-557.
And simply write TNF in place of TNF-α.
L32 The human gastrointestinal tract is a dense microbial habitat hosting approximately 4 × 1013 bacteria, predominantly Firmicutes, Bacteroidetes, Proteobacteria, Actinobacteria, and Verrucomicrobia (Shetty et al., 2019).
The mentioned paper does not specify the approximate number of bacteria, please provide the appropriate reference.
L41 B. fragile
B. fragilis
L48 For instance, supplementation with Lactobacilli has exhibited favorable effects on blood glucose levels…
For instance, supplementation with Lactobacilli (phylum Firmicutes), has exhibited favorable effects on blood glucose levels…
L75 However, some studies have documented the potential pathogenicity of certain BAC strains. Unlike conventional probiotics, the health-promoting attributes of BAC spp. are strain-dependent.
Please provide reference. Some other studies have documented the potential pathogenicity of certain commercial probiotic strains. Please see- Ambesh, P., Stroud, S., Franzova, E., Gotesman, J., Sharma, K., Wolf, L. and Kamholz, S., 2017. Recurrent Lactobacillus bacteremia in a patient with leukemia. Journal of Investigative Medicine High impact case reports, 5(4), p.2324709617744233. Also, the health-promoting properties of conventional probiotics are strain dependent.
Author Response
The paper is well written and well organized. Minor revision is needed, mostly in the introduction.
L13 Bacteroides (BAC) spp.
Bacteroides spp. (BAC)
> We thankfully acknowledge the Reviewer for addressing these vital points. Accordingly, we revised this sentence in the 'Abstract’ as below:
Bacteroides spp. (BAC), which are commensal bacteria, have demonstrated substantial anti-obesity effects in preclinical and clinical studies; however, their underlying biochemical pathways are yet to be fully deciphered.
L16 BAC spp.
Just BAC, remove spp. It needs to be corrected throughout the entire manuscript (L42, L53, L56, L65, L83, L134,166,271, 316…).
> We thankfully acknowledge the Reviewer for addressing these vital points. Accordingly, we have checked the full manuscript and revised the text in its entirety.
e.g.:second, to assess the effectiveness of BAC in combating obesity; and third, to explore the potential mechanisms through which BAC affect obesity and lipid metabolism (L17, L18, L19).
L27 TNF-α
TNF is still referred to as TNF-α in numerous newly published scientific papers, almost 2 decades after the cytokine was renamed. Please see Grimstad, Ø., 2016. Tumor necrosis factor and the tenacious α. JAMA dermatology, 152(5), pp.557-557.
And simply write TNF in place of TNF-α.
> We thankfully acknowledge the Reviewer for addressing these vital points. Accordingly, we revised this sentence in the 'Abstract’ as below:
“Furthermore, BAC administration augments the production of TNF by dendritic cells in response to LPS stimulation, which suggests BAC ingestion as an adjunctive therapy for obesity.”
L32 The human gastrointestinal tract is a dense microbial habitat hosting approximately 4 × 1013 bacteria, predominantly Firmicutes, Bacteroidetes, Proteobacteria, Actinobacteria, and Verrucomicrobia (Shetty et al., 2019).
The mentioned paper does not specify the approximate number of bacteria, please provide the appropriate reference.
> We sincerely apologize for these mistakes and thankfully acknowledge the Reviewer for pointing these errors. Accordingly, We have added the following text to the revision and reference:
- The human gastrointestinal tract is a dense microbial habitat hosting approximately 4 × 1013 bacteria, predominantly Firmicutes, Bacteroidetes, Proteobacteria, Actinobacteria, and Verrucomicrobia (Afrizal et al., 2022; Shetty et al., 2019).
- Afrizal, A., Hitch, T.C., Viehof, A., Treichel, N., Riedel, T., Abt, B., Buhl, E.M., Kohlheyer, D., Overmann, J. and Clavel, T., 2022. Anaerobic single‐cell dispensing facilitates the cultivation of human gut bacteria. Environmental Microbiology 24: 3861-3881.
L41 B. fragile
- Fragilis
> We sincerely apologize for these mistakes and thankfully acknowledge the Reviewer for pointing these errors. Accordingly, we have rectified the spellings as follows:
L41: ‘species such as B. fragilis, B. ovatus, B. eggerthii, B. uniforms, and B. "B5-21" are less abundant but still present in substantial concentrations (109 per g dry weight of feces).’
L48 For instance, supplementation with Lactobacilli has exhibited favorable effects on blood glucose levels…
For instance, supplementation with Lactobacilli (phylum Firmicutes), has exhibited favorable effects on blood glucose levels…
> We sincerely acknowledge the concern of the Reviewer on this point and modified accordingly as follows:
L48: ‘For instance, supplementation with Lactobacilli (phylum Firmicutes), has exhibited favorable effects on blood glucose levels.’
L75 However, some studies have documented the potential pathogenicity of certain BAC strains. Unlike conventional probiotics, the health-promoting attributes of BAC spp. are strain-dependent.
Please provide reference. Some other studies have documented the potential pathogenicity of certain commercial probiotic strains. Please see- Ambesh, P., Stroud, S., Franzova, E., Gotesman, J., Sharma, K., Wolf, L. and Kamholz, S., 2017. Recurrent Lactobacillus bacteremia in a patient with leukemia. Journal of Investigative Medicine High impact case reports, 5(4), p.2324709617744233. Also, the health-promoting properties of conventional probiotics are strain dependent.
> We sincerely acknowledge the concern of the Reviewer on this point. Accordingly, we have rearranged the text as per advice of the Reviewer and included this modified version in the revised manuscript.
L76: However, previous research records show that BAC is identical to some commercial conventional probiotics (Ambesh et al., 2017; Mikucka et al., 2022; Widyastuti et al., 2021). The health-promoting properties also depend on the strain.
L449: Ambesh, P., Stroud, S., Franzova, E., Gotesman, J., Sharma, K., Wolf, L. and Kamholz, S., 2017. Recurrent Lactobacillus bacteremia in a patient with leukemia. Journal of Investigative Medicine High impact case reports 5: 2324709617744233.
L552: Mikucka, A., Deptuła, A., Bogiel, T., Chmielarczyk, A., Nurczyńska, E. and Gospodarek-Komkowska, E., 2022. Bacteraemia Caused by Probiotic Strains of Lacticaseibacillus rhamnosus—Case Studies Highlighting the Need for Careful Thought before Using Microbes for Health Benefits. Pathogens 11: 977.
L621: Widyastuti, Y., Febrisiantosa, A. and Tidona, F., 2021. Health-promoting properties of lactobacilli in fermented dairy products. Frontiers in Microbiology 12: 673890.
Reviewer 3 Report
Comments and Suggestions for Authors
Dear Authors,
Thank you for this manuscript. It is written in an easy to read style with very good English skills. Please find my comments per line below:
Line 53: Can you please give some literature for animal models as Sanchis‑Chorda et al., 2019 worked with children.
Line 62: Please change the . to a ,
Line 84: Could you please start a new paragraph beginning with “In our analysis…”
Line 100: Please give all search terms and filters used. Did you set an earliest publication date?
Line 117: Did you think of Cachexia as marker to use?
Line 126: Here you declare “no language restrictions” but line 101 says “only published in English”. Please clarify!
Table 1/2/3/4/5: The start of the table using names in brackets looks a little bit strange. Could you please name this column and use it officially? Maybe put it as second column to find the strains more easily.
Table 2: Please put Ruminococcus to italic letters; Please put Shigella to italic letters; Please put Faecalibacterium prausnitzii to italic letters; Please put Prevotella stercorea to italic letters;
Line 255: Please set the missing space between table and 3
Table 3: Please put Bifidobacterium to italic letters;
Line 266: Please rephrase “In a 2009 study by”
Table 4: Please put Prevotella to italic letters; Please correct P < 0.0001 to p < .0001;
Line 321: Please explain how you found “at least 6 weeks”. Did you compare studies with shorter application or is there a lack of data and so it is not known if shorter application would work, as far as the described rats study applicated for 18 days with significant results.
Line 364/365ff: Please insert some sentences concerning the ability of humans decide what they eat and animals being fed with the same diet all as possible factor for this.
Study limitations: Please insert the availability of in vivo studies for male animals only.
Line 410: Please set examine to examines
Author Response
Thank you for this manuscript. It is written in an easy to read style with very good English skills. Please find my comments per line below:
Line 53: Can you please give some literature for animal models as Sanchis‑Chorda et al., 2019 worked with children.
> We thankfully acknowledge the Reviewer for addressing these vital points. Accordingly, we revised this sentence in the 'Introduction’ as below:
Line 51: ‘Bifidobacterium spp. has also effectively reduced fat and body weight in preclinical and clinical studies of obesity and type 2 diabetes.(Le et al., 2015; Sanchis-Chorda et al., 2019).’
Line 62: Please change the . to a ,
> We have checked manuscript in line 62 and modified accordingly as follows:
L63: ‘... is a symbiotic factor that stimulates immune development in mammalian hosts, stimulates.....’
Line 84: Could you please start a new paragraph beginning with “In our analysis…”
> We have checked manuscript in line 84 and modified accordingly as follows:
L85: ‘In our analysis, the efficacy of BAC in preventing obesity in humans remains largely unexplored, necessitating further investigation.’
Line 100: Please give all search terms and filters used. Did you set an earliest publication date?
> We sincerely acknowledge the concern of the Reviewer on this point. Accordingly, we have rearranged the text as per advice of the Reviewer and included this modified version in the revised manuscript.
Line 100: We conducted a comprehensive literature search in November 2022 using the PubMed, Google Scholar, Cochrane, Scopus, and Web of Science databases within a restricted time period between January 1, 2000, until November 30, 2022. We employed specific search strings combining terms and Boolean operators, such as “Bacteroides AND obesity AND trial,”. These terms were adapted for each literature database and combined with database-specific filters for human clinical trials and preclinical trials in animals.
Line 117: Did you think of Cachexia as marker to use?
> We sincerely acknowledge the concern of the Reviewer on this point. Cachexia is a progressive metabolic disorder. It can co-exist with obesity, but can also lead to overconsumption of adipose tissue. Therefore, we do not consider Cachexia to be a usable marker after careful consultation.
You can refer to the following references:
- Prado, C. M., et al. "Sarcopenia and cachexia in the era of obesity: clinical and nutritional impact." Proceedings of the Nutrition Society2 (2016): 188-198.
- Rohm M, Zeigerer A, Machado J, et al. Energy metabolism in cachexia[J]. EMBO reports, 2019, 20(4): e47258.
Line 126: Here you declare “no language restrictions” but line 101 says “only published in English”. Please clarify!
> We sincerely apologize for these mistakes and thankfully acknowledge the Reviewer for pointing these errors.
Our search criteria are “no language restrictions”, Accordingly, we have made changes to the original text.
Line 255: Please set the missing space between table and 3
> We sincerely apologize for these mistakes and thankfully acknowledge the Reviewer for pointing these errors. Accordingly, I deleted the spaces.
Table 1/2/3/4/5: The start of the table using names in brackets looks a little bit strange. Could you please name this column and use it officially? Maybe put it as second column to find the strains more easily.
> We thankfully acknowledge the Reviewer for addressing these vital points. Accordingly, After our deliberations, we gave names to the beginning of the tables and made changes in the original text, but tables 2, 3, and 4, since there is no specific BAC strain, did not switch table positions.
Table 2: Please put Ruminococcus to italic letters; Please put Shigella to italic letters; Please put Faecalibacterium prausnitzii to italic letters; Please put Prevotella stercorea to italic letters;
> We sincerely apologize for these mistakes and thankfully acknowledge the Reviewer for pointing these errors. Therefore, I have made the following changes in Table 2:
- ‘Ruminococcus’; ‘Shigella’; ‘Faecalibacterium prausnitzii’; ‘Prevotella stercorea’
Table 3: Please put Bifidobacterium to italic letters;
> We sincerely apologize for these mistakes and thankfully acknowledge the Reviewer for pointing these errors. Therefore, I have made the following changes in Table 3:
- ‘Bifidobacterium’
Line 266: Please rephrase “In a 2009 study by”
> We sincerely acknowledge the concern of the Reviewer on this point. Accordingly, we have rearranged the text as per advice of the Reviewer and included this modified version in the revised manuscript.
Line 270: A survey of the gut microbiota of 46 pairs of twins and their mothers revealed. The obese gut microbiota was significantly different (P = 0.003) and less diverse than the lean core gut microbiome (Turnbaugh et al., 2009).
Table 4: Please put Prevotella to italic letters; Please correct P < 0.0001 to p < .0001;
> We sincerely apologize for these mistakes and thankfully acknowledge the Reviewer for pointing these errors. Accordingly, I have made the following changes in Table 4:
- ‘Prevotella; p < .0001’
Line 321: Please explain how you found “at least 6 weeks”. Did you compare studies with shorter application or is there a lack of data and so it is not known if shorter application would work, as far as the described rats study applicated for 18 days with significant results.
> We sincerely acknowledge the concern of the Reviewer on this point.
While 18 days can affect total caloric intake, the direct effect on body weight is not significantly different, so I chose 7 weeks as the minimum gavage duration.
we have rearranged the text as per advice of the Reviewer and included this modified version in the revised manuscript.
Line 326: In our analyses, the BAC measure had to be at least 1×106 CFU/day and lasted at least 7 weeks to present validity in the body weight results.
Line 364/365ff: Please insert some sentences concerning the ability of humans decide what they eat and animals being fed with the same diet all as possible factor for this.
> We sincerely acknowledge the concern of the Reviewer on this point. Accordingly, we have rearranged the text as per advice of the Reviewer and included this modified version in the revised manuscript.
Line 371: There are also studies that show, human dietary preferences in animal husbandry practices may influence domesticated animals (Reese et al., 2021). Dietary homogenization among different species may lead to convergence of microbial characteristics.
Line 570: Reese, A.T., Chadaideh, K.S., Diggins, C.E., Schell, L.D., Beckel, M., Callahan, P., Ryan, R., Emery Thompson, M. and Carmody, R.N., 2021. Effects of domestication on the gut microbiota parallel those of human industrialization. Elife 10: e60197.
Study limitations: Please insert the availability of in vivo studies for male animals only.
> We sincerely acknowledge the Reviewer's concern on this point. Cachexia is a progressive metabolic disorder. The body weight whole-body fat content of males was higher than that of females, and protection of female mice from obesity appears to depend on diet and animal strain. That is why most preclinical experiments on obesity use male animals.
You can refer to the following references:
- Mestdagh, R., Dumas, M.-E., Rezzi, S., Kochhar, S., Holmes, E., Claus, S.P. and Nicholson, J.K., 2012. Gut microbiota modulate the metabolism of brown adipose tissue in mice. Journal of proteome research 11: 620-630.
Line 410: Please set examine to examines
> We have checked manuscript in line 410 and modified accordingly as follows:
Line 418: ‘...that directly examines the.....’
> We appreciate your valuable feedback.
Round 2
Reviewer 1 Report
Comments and Suggestions for Authors
Thank you to the authors for sharing their comments. The manuscript now reads more smoothly and appears to be more organized. In spite of the fact that the authors have stated that the references comply with the guidelines, they may wish to check in the PDF names rather than numbers.
Author Response
Thank you to the authors for sharing their comments. The manuscript now reads more smoothly and appears to be more organized. In spite of the fact that the authors have stated that the references comply with the guidelines, they may wish to check in the PDF names rather than numbers.
> We greatly appreciate your insightful and understand your concern. In response to your valuable suggestion, we changed the PDF names to numbers according to the guidelines for the references.
In addition, we have revised abstract to reduce the words in compliance with the guidelines.